# The Influence of Maturity Status on Dynamic Balance Following 6 Weeks of Eccentric Hamstring Training in Youth Male Handball Players

**DOI:** 10.3390/ijerph19159775

**Published:** 2022-08-08

**Authors:** Ammar Nebigh, Raouf Hammami, Sofien Kasmi, Haithem Rebai, Benjamin Drury, Mokhtar Chtara, Roland van den Tillaar

**Affiliations:** 1Higher Institute of Sport and Physical Education of Ksar-Said, University of La Manouba, Tunis 2010, Tunisia; 2Research Laboratory: Education, Motor Skills, Sports and Health (LR19JS01), Higher Institute of Sport and Physical Education of Sfax, University of Sfax, Sfax 3018, Tunisia; 3Department of Applied Sport Sciences, Hartpury University, Gloucester GL19 3BE, UK; 4Tunisian Research Laboratory, Sport Performance Optimization, National Center of Medicine and Science in Sports (CNMSS), Tunis 1004, Tunisia; 5Department of Sports Science, Nord University, 7600 Levanger, Norway

**Keywords:** peak height velocity, dynamic balance, neuromuscular performance, youth training, eccentric resistance training

## Abstract

Information about when to apply an appropriate eccentric hamstring strength training stimulus during long-term athlete development is essential for effective programming and improving balance performance. This study examined the effects of six-week eccentric hamstring training on dynamic balance performance in youth handball players of different maturity statuses (pre- or post-peak height velocity (PHV)). A randomized controlled design with 45 young male handball players (14.6 ± 0.3 years) from a local national handball club were randomly allocated to a 6 week, twice weekly eccentric hamstring training in two experimental groups: a pre-PHV (maturation offset: −2.13 ± 0.63, *n* = 10) and post-PHV (maturation offset: 0.79 ± 0.34, *n* = 12) group and two control groups: maturation offset: −2.09 ± 0.61, *n* = 10 and maturation offset: 0.55 ± 0.67, *n* = 13. Dynamic balance performance was evaluated by using the composite score during the lower quarter Y-balance test from pre- and post-intervention. A significant effect on balance scores was found from pre to post (F = 11.4; *p* = 0.002; η^2^_p_ = 0.22), intervention (F = 5.4; *p* = 0.025; η^2^_p_ = 0.12), and maturation (F = 369; *p* < 0.001; η^2^_p_ = 0.9), but no significant interaction effects were found (F ≥ 3.3; *p* ≥ 0.077; η^2^_p_ ≤ 0.07). Post hoc analysis revealed that the post-PHV group had a higher score than the pre-PHV group. Furthermore, that dynamic balance increased in the post-PHV group after intervention much more in the control post-PHV group. It was concluded that somatic maturation influences dynamic balance performance and that eccentric hamstring training results in greater improvement in balance performance in young male handball players for the post-pubertal group.

## 1. Introduction

Technical skills in handball such as jumping and pivoting activities have been considered as injury risk factors [1] with many studies proposing prevention purposes based on improving athletes’ balance and stability [2,3,4]. Different studies have indicated that ballistic strength [5], eccentric strength [6,7], plyometric [8], and balance [9] training improved dynamic balance performance. While the training effects of these training modalities are similar on athletic performance, they differ according to their neural control mechanisms [10,11]. This is particularly pertinent to youth athletic populations as maturation status is known to influence training adaptations [12,13]. Moreover, training of balance throughout all maturity stages is recommended to support long-term athlete development [14]. Therefore, knowledge of training modalities that may enhance balance in youth athletes can provide practitioners with helpful information to improve performance and reduce injury risk.

While human movements in youth athletes will naturally include activities such as controlling balance during jumping, landing, and hopping, performing eccentric muscle exercise may already occur within the demands of sport or even playground actions [15]. Like in running during the late swing phase, the biceps femoris and hamstring muscles eccentrically resist the hip flexion and decelerate knee extension [16], where they undergo large ranges of motion and different activation patterns. It would therefore be advisable to train the hamstrings eccentric with both hip and knee dominant exercises. Considering the above, Drury et al. [17] examined the effects of a 6-week Nordic hamstrings exercise training in youth male soccer players of pre or mid and post-PHV athletes and reported improvements in eccentric hamstring strength performance in both groups.

Since balance and muscle strength are related in youth with a less developed neuromuscular system [18], improvement in balance performance was considered as a good tool for young handball players as they must enhance coordination during dribbling, passing and throwing the ball, while sprinting and jumping during the offensive and defensive actions [19]. Furthermore, major morphological and neural changes occur with growth and maturation [20]. These asynchronously changing parameters in youth play an important role in the ability to adapt to an eccentric resistance training stimulus [17,21]. Hence, knowledge of when to apply an appropriate eccentric resistance training stimulus during long-term athletic development for youth handball players is important for good programming and improving postural control performance. In this context, to estimate the maturity status of participants, a maturity index (i.e., timing of maturation) could be calculated [18]. This assessment is a noninvasive and practical method of predicting years from peak height velocity (PHV) as a measure of maturity offset using height and age as variables. Furthermore, individual physiological changes (e.g., hormonal, central nervous system myelinization) may favor different types of adaptation depending on maturity status. The importance of eccentric muscular strength for long-term athletic development for performance enhancement have recently been described by Hammami, Duncan, Nebigh, Werfelli, and Rebai [7]. Particularly, dynamic balance is an important prerequisite for motor skill acquisition and motor performance enhancement [18].

Therefore, the objective of the present study was to investigate the effects of a six-week in-season eccentric hamstring training on dynamic balance performance in male athletes at different maturity statuses (pre- and post-PHV). Based on the available longitudinal [13] and meta-analytical [12] studies, we hypothesized that the post-PHV players would respond more to the eccentric training hamstring strength.

## 2. Materials and Methods

### 2.1. Study Design

This study used a repeated measures design in which participants with different maturation levels undertook the eccentric hamstring training program in five distinct exercises, while the participants were the control group. The dependent variable was the balance test which was tested before and after the six-week training period.

### 2.2. Participants

A minimum sample size of 25 was determined from an a priori statistical power analysis using G*Power (Version 3.1, University of Dusseldorf, Düsseldorf, Germany) [22]. The power analysis was computed with an assumed power at 0.90 at an alpha level of 0.01 and a moderate effect size of 0.36 for our primary outcome Y-balance test [9]. Forty-five, young, male handball players, aged between 9 and 14 years, participated in the study (see Table 1). All subjects were free from any musculoskeletal injuries, and they were physically active, and participated regularly in handball practice. All subjects performed a systematic handball training in the first division of the Tunisian national handball league for at least 2–4 years. All participants performed a minimum of five times per week with each session lasting ~90 min and one match played during the weekend. Subjects were not undertaking any further activities other than the team handball training.

Before the study, all subjects were given a letter including written information about the study and a consent from the parents to allow their children to participate in the study. Legal representatives and players provided informed consent after thorough explanation of the objectives and scope of this project, the procedures, risks, and benefits of the study. The study was conducted according to the latest version of the Declaration of Helsinki and the protocol was fully approved by the Local Ethics Committee of the National Centre of Medicine and Science of Sports of Tunis (CNMSS-LR09SEP01) before the commencement of the procedure.

### 2.3. Procedure

The intervention was conducted during the second half of the season (April–June 2020). Before the commencement of the study and prior to the assessment, all players completed a two-week orientation period (three sessions/week) to become familiar with the general environment, form, and technique of the Y-Balance test. During this period, all participants received instructions on proper technique for the eccentric hamstring exercises from certified strength and conditioning specialists. The training routine comprised five repeated ~ 90 min training sessions with a competitive game played on Sunday. Training consisted mainly of tactical skill development (60% of session time) and strength and conditioning routines (40% of session time). Anthropometric parameters were assessed prior to balance testing. Each player’s height and body mass were assessed using a wall-mounted stadiometer (i.e., OHAUS, Florhman Park, NJ, USA) and an electronic scale (i.e., Baty International, West Sussex, UK), respectively. The sum of skinfolds was monitored with Harpenden skinfold callipers (Baty International, West Sussex, UK). Body measurements were conducted according to Deurenberg et al. [23], who reported similar prediction errors between adult and young populations. Leg length was measured from the anterior superior iliac spine to the mid-medial malleolus until 3 consecutive measurements were the same [24]. Thereafter, biological maturity was assessed non-invasively by incorporating measures of chronological age and body height into a regression equation able to predict biological age from PHV [25]. Participants were allocated to two groups according to their maturity status. To estimate the maturity status of participants, a maturity index (i.e., timing of maturation) was calculated [25]. This assessment is a noninvasive and practical method of predicting years from peak height velocity (PHV) as a measure of maturity offset using height and age as variables (PHV = 27.999994 + [0.0036124 × age × height]). In general, participants can be classified into three categories according to their maturity status: pre-PHV velocity (−3 years to >−1 years from PHV) and post-PHV (>1 to +3 years from PHV) [25]. The equation has previously been validated with youth athletes with standard error of estimates reported as 0.57 and 0.59 years, respectively [18].

#### 2.3.1. Dynamic Balance

Balance performance was assessed using the Y-Balance Test according to a previously described protocol [21]. The protocol used for the completion of the Y-balance test has been reported to possess high reliability in youth athletes [26]. For this purpose, players stood on the dominant leg, with the most distal aspect of their big toe on the center of the grid. Thereafter, they were asked to reach the maximal distance in the anterior (A), postero-medial (PM), and postero-lateral (PL) directions, while maintaining their single-limb stance (Fusco et al., 2020). The maximal reached distance was measured with a measuring tape as the most distal point reached by the free limb. The trial was discarded and repeated if the player failed to maintain a unilateral stance, touched down with the reaching foot, or failed to return the reaching foot to the starting position. A composite score [CS-YBT (%)] was calculated and considered as the dependent variable using the following formula: CSYBT (%) = [(maximum anterior reach distance + maximum posteromedial reach distance + maximum posterolateral reach distance)/(leg length × 3)] × 100.

#### 2.3.2. Training Program

After the pretest, somatic maturity was predicted from measures of chronological age and body height to predict biological age from PHV [12]. The equation has previously been considered valid for boys and presents a standard error of the estimate reported as 0.542 years [27]. Therefore, a maturity offset of −1.0 indicates that the participant was measured 1 year before this peak velocity; a maturity offset of 0 and +1.0 indicates that the subject was measured after this peak velocity [28]. Accordingly, the players were stratified randomly allocated into two experimental groups: a pre-PHV (maturation offset: −2.13 ± 0.63, *n* = 10) and Post-PHV (maturation offset: 0.79 ± 0.34, *n* = 12) group and two control groups: maturation offset: −2.09 ± 0.61, *n* = 10; maturation offset: 0.55 ± 0.67, *n* = 13.

The training program consisted of a progressive 6-week eccentric hamstring training program (Table 2). All participants performed a standardized warm up consisting of low-intensity aerobic, agility, plyometric, and dynamic lower-limb stretching exercises. After that, participants of the intervention group were familiarized to the eccentric hamstring training program. The eccentric hamstring training was performed before the regular handball training routine. Participants were considered ‘familiarized’ when they could perform multiple repetitions with the correct technique. Participants did not have any symptoms of exercise-induced injuries (e.g., reduced muscle function or elevated muscle soreness) throughout the study. The eccentric hamstring training program consisted of two sessions per week with 3–5 sets per session and 10–12 repetitions per set (Table 2). The design of the program was defined from previous recommendations with the volume of Nordic hamstring exercise training progressively increasing weekly [27,29]. Identical weekly increases in training load were reported for both experimental groups [7,30]. Whilst the experimental group performed the training program, the control group executed a low intensity passing drills until the main training session began in which both groups completed the same handball training. The between-session recovery time was 48 h.

### 2.4. Statistical Analysis

Means ± standard deviations (SDs) were used to describe variables. The Shapiro–Wilk and Mauchly tests confirmed the data’s normal distribution and sphericity, respectively. An independent sample t-test between the intervention and control groups on each of the anthropometric variables at the pretest was performed. The balance data was analyzed using mixed model 2 (time: pre–post; repeated measures) × 2 (group: experimental and control) × 2 (pre–post PHV group) ANOVA. Effect size was evaluated with η^2^ (partial eta square) where 0.01 < η^2^_p_ < 0.06 constitutes a small effect, 0.06 < η^2^_p_ < 0.14 constitutes a moderate effect, and η^2^_p_ > 0.14 constitutes a large effect [30]. Statistical analyses were performed using SPSS (SPSS Inc., Chicago, IL, USA, version 28.0), and significance was accepted, a priori, at *p* < 0.05.

## 3. Results

Adherence rate was 100% across all groups and none reported any training- or test-related injury. No significant differences in anthropometric characteristics were found between the control and intervention groups (F ≤ 6.5, *p* ≥ 0.20, η^2^_p_ ≤ 0.03, Table 1). When comparing the effect of the intervention and PHV maturation a significant effect on balance scores was found from pre to post (F = 11.4; *p* = 0.002; η^2^_p_ = 0.22), intervention (F = 5.4; *p* = 0.025; η^2^_p_ = 0.12), and maturation (F = 369; *p* < 0.001; η^2^_p_ = 0.9), but no significant interaction effects (F ≥ 3.3; *p* ≥ 0.077; η^2^_p_ ≤ 0.07). Post hoc comparisons revealed that the post-PHV group had a higher score than the pre-PHV group. Furthermore, in the pre-PHV none of the groups (control and intervention) increased their balance scores significantly and had similar scores at pre- and post-test, while in the post-PHV group the control group had a significantly lower balance score than the intervention group at the pretest and post-test. In addition, in the post-PHV group both groups increased their balance scores from pre to post test. However, the intervention group increased the score significantly (13.5 vs. 2.8%) more than the control group (Figure 1).

## 4. Discussion

The aim of the present study was to investigate the effects of eccentric hamstring training on balance performance in young male handball players of different maturity statuses. The main findings of the current study were that responses to eccentric hamstring training in youth handball players of different maturity status appear to be different for improving dynamic balance in youth. Both post-PHV groups (control and intervention) showed improvements in balance, but the intervention post-PHV group had a higher increase in balance than the control group (13.5 vs. 2.8%), while no significant changes in balance were found in the pre-PHV group following 6 weeks of training.

Only in the post-PHV group were increases observed in both groups, with a significantly higher increase in the intervention group. Balance performance in the present study was improved to a greater extent with eccentric training in the post-PHV young handball players compared to the control group. Thus, with regard to the principle of training specificity [31] and with the added stress of a specific resistance, using eccentric exercise provides considerable tension training stress to the participant that could enhance balance performance. The latter could be explained by the fact that better stability enhancement was reasoned to the muscle adaptation towards the strengthening exercises, either neural or structural adaptation [32]. In this context, the structure of muscle adaptation as a result from the overload resistance may consist of muscle fiber hypertrophy and hyperplasia. The neural factors influence strength performance by increased neuron firing recruitment, improved motor unit synchronization and coordination, as well as boosted agonist muscles’ activity, decreased antagonist muscle activity, and inhibited muscle protection mechanism (Golgi tendon organ) [33].

Furthermore, the pre-PHV groups demonstrated a lower score of dynamic balance compared to post-PHV groups. In addition, a lower score at pretest for the control group, than the intervention group, was observed. It was considered that balance and stability are not mature in youth because their neuromuscular system is not fully developed and their motor skills are still emerging [18,34]. Postural control improves with maturation [35] and thus enhanced coordination would contribute to improved balance and stability around PHV [36]. Hence, the better balance performance by the post-PHV group would be expected. This finding might be explicated by the fact that the rapid growth of the PHV phase constantly modifying the center of gravity provides novel challenges to the better postural control for this group. The vestibular systems of the post-PHV groups would have had a more consistent morphology (i.e., height, mass, limb, and trunk lengths) over time with which to optimize their strategies to maintain static balance more after than the pre-PHV group.

Furthermore, the adaptive processes obtained after eccentric training exercises may contribute to improved hamstring activation during standing balance, which prepares the muscles to exert an optimal hip-extensor moment during standing balance. More increase in balance was found for the post-PHV group compared with the pre-PHV group. The latter could be explained by the fact that the fast contraction velocities during eccentric hamstring training may provide an augmented training response, recently termed “synergistic adaptation” [13,37], that can generate greater improvement in balance performance in post-PHV more after than the pre-PHV group. In addition, post-PHV players experience morphological changes that facilitate force production (e.g., increased motor unit size and pennation angles) to continued neural adaptations because of cognitive maturation [20]. Furthermore, it is important to note that with handball training experience of 2–4 years, the post-PHV groups would have a prior greater experience of resistance training. As there was a difference in balance performance between maturation groups, further studies may wish to investigate the effects of different eccentric training volumes or frequencies in youth athletes to ensure that training can be optimized in both pre- and post-PHV players.

This study is not without limitations. Firstly, we examined a sample of youth handball players. Therefore, the results of this study are specific to the population under investigation. Furthermore, the balance test does not compensate for limb length and thereby favors taller players when comparing different age groups. Thus, perhaps the test should be better when compensating for limb length. Further studies should focus on these aspects. Secondly, the protocol execution for assessment of dynamic balance performance (i.e., YBT) has a possible limit, that is, it does not compensate for the height of the pelvis of the youth participant, so all participants will certainly have a higher value, which does probably affect the results. Therefore, future studies are advised to include balance testing apparatus (e.g., stabliometric force plate). Balance performance was therefore tested in bipedal stance on an unstable surface (i.e., BOSU ball with flat side facing up) using two dependent variables, i.e., center of pressure surface area (CoP SA) and velocity (CoP V). Lastly, balance performance was not tested under sport-specific conditions in this study, which may have prevented any observation of larger positive effects. Further studies should include balance tests during the performance of handball activities. Players could therefore execute a ball shooting technique or block technique whilst jumping from and landing on a force plate.

## 5. Conclusions

The results of this study demonstrate that eccentric training twice weekly for six weeks results in improvements in balance performance in male handball players. However, maturation seems to affect balance performance enhancement from eccentric training with the post-PHV group obtaining greater improvements than the pre-PHV group. Using eccentric hamstring training in the current study is therefore a good tool for coaches and practitioners to enhance in-season balance performance in both groups with a greater improvement in the more mature group (post-PHV). Enhanced in balance performance was considered as a good tool for young handball players as they must enhance coordination during dribbling, passing, and throwing the ball, while sprinting and jumping during the offensive and defensive actions. Consequently, the training program utilized within this study may help practitioners in the field to implement the eccentric hamstring training in addition to their training programs for injury prevention purposes.

## Figures and Tables

**Figure 1 ijerph-19-09775-f001:**
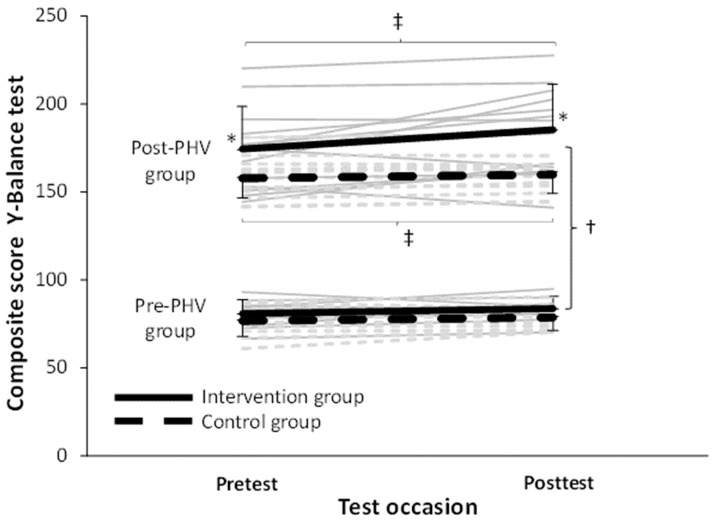
Individual score in the balance test at pre- and post-test for each participant together with mean (SD) of intervention and control group of the pre- and post-PHV groups at pre- and post-tests. * indicates a significantly higher score than control group at this test on a *p* < 0.05 level. † indicates a significant difference between pre- and post-PHV groups on a *p* < 0.05 level. ‡ Indicates a significant increase from pre to post test for this group on a *p* < 0.05 level.

**Table 1 ijerph-19-09775-t001:** Mean ± SD of anthropometric characteristics of the groups and difference between groups measured with a one-way ANOVA (control vs. intervention group) with effect size (η^2^_p_) at the pretest.

	Maturity	Intervention Group	Control Group	*p*-Value	η^2^_p_
**Age** (**years**)	**Pre-PHV**	11.24 ± 0.93	11.03 ± 0.77	0.46	0.01
	**Post-PHV**	14.00 ± 0.20	13.85 ± 0.87		
**Height** (**cm**)	**Pre-PHV**	144.20 ± 4.16	147.80 ± 8.36	0.73	0.01
	**Post-PHV**	173.50 ± 61.03	170.85 ± 4.81		
**Sitting height** (**cm**)	**Pre-PHV**	72.00 ± 3.89	72.70 ± 4.67	0.88	0.01
	**Post-PHV**	50.38 ± 3.15	50.38 ± 3.15		
**Body mass** (**kg**)	**Pre-PHV**	37.60 ± 5.32	42.10 ± 6.23	0.99	0.01
	**Post-PHV**	67.83 ± 13.86	65.23 ± 7.41		
**BMI** (**kg/m^2^**)	**Pre-PHV**	17.98 ± 1.87	19.29 ± 2.19	0.20	0.04
	**Post-PHV**	22.56 ± 4.77	20.79 ± 4.20		
**PHV**	**Pre-PHV**	−2.13 ± 0.63	−2.09 ± 0.61	0.61	0.01
	**Post-PHV**	0.79 ± 0.34	0.55 ± 0.67		
**Age PHV**	**Pre-PHV**	13.38 ± 0.33	13.12 ± 0.38	0.25	0.03
	**Post-PHV**	14.79 ± 0.44	14.40 ± 1.52		

PHV: peak height velocity; BMI: body mass index.

**Table 2 ijerph-19-09775-t002:** Design of the training program.

Exercises	Duration (s)	Week					
Load		1	2	3	4	5	6
Nordic hamstring exercise	3–5	3 × 10	4 × 12	5 × 12	3 × 10	4 × 12	5 × 12
Manual glute–hamstring rise	3–5	3 × 10	4 × 10	5 × 12	3 × 12	3 × 10	5 × 12
Single leg Romanian dead lift	3–5	3 × 10	4 × 12	5 × 12	3 × 10	4 × 12	5 × 12
Glute bridge and hip thrust free weight progressed to dumbbell and barbell	3–5	3 × 10	4 × 12	5 × 12	3 × 10	4 × 12	5 × 12
Good morning dumbbell or barbell	3–5	3 × 10	4 × 12	5 × 12	3 × 10	4 × 12	5 × 12

## Data Availability

The data presented in this study are available on reasonable request from Dr. Ammar Nebigh. Requests for access to data should be sent to ammarnebigh612@gmail.com.

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
