# Peer review of "The Influence of Maturity Status on Dynamic Balance Following 6 Weeks of Eccentric Hamstring Training in Youth Male Handball Players"

_ijerph, 2022, doi:10.3390/ijerph19159775_

Round 1

Reviewer 1 Report

Review

Abstract: Begin abstract with some introduction of the topic area

Introduction:

Line 57: PHV used without explanation of the abbreviation.

Overall: Explain the choice of PHV as the dividing line between maturity levels. How and why is this metric used? How is it calculated.

Can PHV be determined prior the participant

Methods

Figure on line 138: Add caption that explains what statistical test was done. 1-way anova between all ?

Author Response

We want to thank the reviewer for reviewing the manuscript. We have made the changes according to the comments of the reviewer and colored the changes red in the manuscript.

Reviewer 1

Abstract: Begin abstract with some introduction of the topic area

As recommend we begin the abstract section with the introduction statement as follow:

«Information about when to apply an appropriate eccentric hamstring strength training stimulus during long-term athlete development is essential for effective programming and improving balance performance.»

Introduction:

Line 57: PHV used without explanation of the abbreviation.

In line 20 Peak height velocity was written before abbreviation

Overall: Explain the choice of PHV as the dividing line between maturity levels. How and why is this metric used? How is it calculated.

The following statement was added to the method and procedure section to more explain the choice of PHV as the dividing line between maturity levels and how and why is this metric used and calculated:

«Participants were allocated to two groups according to their maturity status. To estimate the maturity status of participants, a maturity index (i.e., timing of maturation) was calculated (Moore et al., 2015). This assessment is a noninvasive and practical method of predicting years from peak height velocity (PHV) as a measure of maturity offset using height and age as variables (PHV = 27.999994 + [0.0036124 × age × height]). In general, participants can be classified into three categories according to their maturity status: pre-PHV velocity (-3 years to > -1 years from PHV) and post-PHV (>1 to +3 years from PHV) (Moore et al., 2015). The equation has previously been validated with standard error of estimates reported as 0.57 and 0.59 years, respectively (Hammami et al., 2016).»

References used:

  • Moore, SA, McKay, HA, Macdonald, H, Nettlefold, L, BaxterJones, ADG, Cameron, N, and Brasher, PMA. Enhancing a somatic maturity prediction model. Med Sci Sports Exerc 47: 1755–1764, 2015.
  • Hammami, R.; Chaouachi, A.; Makhlouf, I.; Granacher, U.; Behm, D. G., Associations between balance and muscle strength, power performance in male youth athletes of different maturity status. Pediatric Exercise Science 2016, 28, (4), 521-534.

Can PHV be determined prior the participant

According to the above, after that participants were recruited and assessed for chronological age and body height measurement, all players were split in to two groups according the formula: (PHV = 27.999994 + [0.0036124 × age × height]). Subsequently, participants can be classified into two categories groupd according to their maturity status: pre-PHV velocity (-3 years to > -1 years from PHV) and post-PHV (>1 to +3 years from PHV) (Moore et al., 2015). Thank you.

Methods

Figure on line 138: Add caption that explains what statistical test was done. 1-way anova between all ?

We have included this to the legend now.

Reviewer 2 Report

The article proposed is a well written manuscript, that aimed to evalued the effects of 6 weeeks training program on dynamic balance performance in handball players at different maturity status.

The article as a whole is well-written, however changes are needed to make it clearer and more fluid to the reader.

Minor comments:

 - Abstract: authors should write the abstract in order to give a very brief overview before moving on to the purpose of the article;

 - Introduction: it would be appropriate to specify what is meant by maturity status and the effects that this condition can generate;

 -Materials: a clarification is necessary, the test with which the balance was assessed has a limit, that is, it does not compensate for the height of the pelvis, so a tall athlete will certainly have a higher value, this does not affect the results?

Author Response

See file
